# Genetic Diversity of Cytochrome P450s *CYP6M2* and *CYP6P4* Associated with Pyrethroid Resistance in the Major Malaria Vectors *Anopheles coluzzii* and *Anopheles gambiae* from Yaoundé, Cameroon

**DOI:** 10.3390/genes14010052

**Published:** 2022-12-23

**Authors:** Yvan Fotso-Toguem, Billy Tene-Fossog, Leon M. J. Mugenzi, Murielle J. Wondji, Flobert Njiokou, Hilary Ranson, Charles S. Wondji

**Affiliations:** 1Department of Medical Entomology, Centre for Research in Infectious Diseases, Yaoundé P.O. Box 13591, Cameroon; 2Department of Animal Biology and Physiology, Faculty of Science, University of Yaoundé 1, Yaoundé P.O. Box 812, Cameroon; 3Department of Vector Biology, Liverpool School of Tropical Medicine, Pembroke Place, Liverpool L35QA, UK

**Keywords:** *CYP6M2*, *CYP6P4*, polymorphism, metabolic resistance, pyrethroid, *Anopheles coluzzii*, *Anopheles gambiae*

## Abstract

Assessing the genetic diversity of metabolic resistance genes, such as cytochrome P450s, helps to understand the dynamics and evolution of resistance in the field. Here, we analyzed the polymorphisms of *CYP6M2* and *CYP6P4,* associated with pyrethroid resistance in *Anopheles coluzzii* and *Anopheles gambiae*, to detect potential resistance markers. Field-caught resistant mosquitos and susceptible lab strains were crossed, and F4 was exposed to permethrin for 15 min and 90 min to discriminate highly susceptible (HS) and highly resistant (HR) mosquitos, respectively. Significant permethrin mortality reduction was observed after pre-exposure to PBO, suggesting the gene involvement of P450s. qPCR analysis revealed significant overexpression of *CYP6M2* (*FC =* 19.57 [95% CI 13.96–25.18] for *An. coluzzii*; 10.16 [7.86–12.46] for *An. gambiae*) and *CYP6P4* (*FC =* 6.73 [6.15–7.30] *An. coluzzii*; 23.62 [26.48–20.76] *An. gambiae*). Full-gene and ≈1 kb upstream were sequenced. For *CYP6M2,* the upstream region shows low diversity in HR and HS (overall Hd = 0.49, π = 0.018), whereas the full-gene shows allelic-variation but without evidence of ongoing selection. *CYP6P4* upstream region showed a lower diversity in HR (Hd = 0.48) than HS (Hd = 0.86) of *An. gambiae*. These results highlighted that *CYP6P4*-associated resistance is potentially driven by modification in upstream region. However, further work is needed to determine the real causative variants that will help design rapid detection tools.

## 1. Introduction

Malaria cases significantly decreased from 2000 to 2015, mainly due to the scaling up of vector control interventions. This decrease was attributed largely to the enormous distribution of Insecticidal Treated Nets (ITNs) with the number of households in sub-Saharan Africa having at least one net increasing from 5% in 2000 to 65% in 2020 [1,2]. Unfortunately, the indiscriminate use of pesticides in the agricultural sector combined with indoor residual spraying (IRS) and the massive distribution of pyrethroid-impregnated nets selected resistance in vector populations in endemic areas [3,4,5,6]. Moreover, the COVID-19 pandemic disrupted vector control activities in most countries, leading to a rebound in mortality from 409,000 malaria associated deaths in 2019 to 627,000 in 2020 [2]. Insecticide resistance, which now extends to almost all classes of insecticides used in public health, remains a major concern for the effectiveness of vector control tools [7]. It is therefore imperative to implement appropriate insecticide resistance management strategies as a proactive approach to maintain the effectiveness of these interventions and to reduce the negative impact of resistance.

The development of insecticide resistance is a complex process depending directly on genetic, physiological, behavioral and ecological factors [8]. In *Anopheles* mosquitos, resistance is mainly conferred by target site and metabolic mechanisms [6,9]. Metabolic resistance involves an increase in production, or more efficient forms, of detoxication enzymes capable of breaking down the insecticide before it reaches its target; the cytochrome P450 family is commonly considered the most important in insecticide detoxification, especially for the metabolism of pyrethroid [9,10]. Cytochrome P450-mediated resistance can often be reversed by the use of synergists, such as piperonyl butoxide (PBO), which inhibit the activity of this enzyme family. Large-scale field trials by Protopopoff et al. [11] and Staedke et al. [12] have shown that nets incorporating this synergist are effective against P450-mediated resistance. Given the continued importance of pyrethroids in ITNs, it is imperative to understand the genetic mechanisms that impact their efficacy to optimize the deployment of PBO-nets and to establish good insecticide resistance management protocols.

Members of the CYP family, including *CYP6M2* and *CYP6P4*, have been functionally associated with metabolic resistance to at least one class of insecticides widely used in vector control, and both are pyrethroid metabolizers [13,14]. *CYP6M2* is overexpressed in multiple populations resistant to type I and II pyrethroids [13,15], organochlorines [16] and carbamates [17]. Interrogation of the *Anopheles gambiae* 100 Genomes project (Ag1000G) identified several amino acid substitutions within *CYP6M2*, grouping into 5 haplotype groups, but none of these was directly linked with resistance [18]. The *CYP6P4* gene has been established as a resistance-associated gene that explains the resistance to permethrin observed in the field in *An. arabiensis* populations in the Sudano–Sahelian region [14,19]. Recently, Njoroge et al. [20] identified in *An. gambiae* a selective sweep localized within a cluster of P450 genes including *CYP6P4* in mosquitos from Uganda. A haplotype, containing three mutations, I236M in *CYP6P4*, an upstream transposable element (TE) insertion and a *CYP6AA1* duplication is strongly associated with pyrethroid resistance in *An. gambiae* from Uganda. In particular, the I236M mutation is strongly associated with increased overexpression of the *CYP6P4* gene capable of metabolizing pyrethroids, notably deltamethrin. Moreover, the overexpression of a group of genes, including *CYP6P4* in *Anopheles* populations, was associated with the loss of efficacy of some ITNs in Tanzania [21].

In Cameroon, P450s have been found widely implicated in metabolic resistance to pyrethroids, DDT and Bendiocarb in *An. gambiae* s.l. populations [22,23,24] and are at least partially responsible for the reduced efficacy of bed nets for *An. gambiae* s.s. [25]. The challenge in detecting simple point-mutation markers for metabolic resistance in contrast to target-site resistance, such as knockdown resistance (kdr), includes the size of the gene families involved in the detoxication process, the redundancy between their members and the multiple mechanisms by which metabolic resistance can arise [26]. A major milestone was recently reached with the development of field-applicable diagnostic tools for P450-mediated resistance in *An. funestus*, taking advantage of a mutation linked to the overexpression of *CYP6P9a* and the variants in *cis*-regulatory elements of *CYP6P9b* [27,28,29]. Such tools are less readily available for *An. gambiae* although, as described above, a triple mutant haplotype in *CYP6P4* was recently linked to resistance in East Africa [20].

Transcriptomic analysis in field populations of *An. gambiae*, *An. coluzzii* and *An. arabiensis* have revealed several overexpressed detoxication genes, irrespective of the presence of kdr-mutations [4,17,22,30]. Similarly, studies conducted by the Ag1000G have identified strong selective sweeps in clusters of *CYPs* known to be implicated in resistance [31]. The elucidation of the roles played by major resistance genes and the changes in the genetic diversity underlying their expression are essential to design DNA-based diagnostic tools to detect and track metabolic resistance in the field.

In this study, we analyzed the polymorphism of *CYP6M2* and *CYP6P4* in *An. gambia*e and *An. coluzzii* to detect potential resistance markers following the same approach as Weedall et al. [29] based on target DNA-sequencing of highly permethrin-resistant and highly susceptible samples selected from reciprocal crosses. We report a comprehensive analysis of the putative promoter and full-gene region of these genes in *An. coluzzii* and *An. gambiae* from Yaoundé-Cameroon.

## 2. Materials and Methods

### 2.1. Ethical Clearance

The National Ethics Committee for Health Research of Cameroon approved the protocol of the study under the ethical clearance No. 2019/10/1195/CE/CNERSH/SP and reference No. 0977/MINSANTE/SESP/SG/DROS/.

### 2.2. Mosquito Samples Processing

#### 2.2.1. Mosquito Collection

Mosquito collections were carried out from December 2018 to March 2019 (dry season) and from July to August 2019 (rainy season) in Ngousso (3°54′22.36″ N, 11°32′20.36″ E) and Nkolondom (3°57′6.99″ N, 11°29′51.67″ E), two neighborhoods of Yaoundé, the capital city of Cameroon. These collection areas have been chosen based on species distribution and the detection of pyrethroid resistance as described in previous studies [22,32,33] predicting the presence of *An. coluzzii* in Ngousso and *An. gambiae* in Nkolondom, with detected metabolic resistance. Larvae were sampled by dipping methods [34], in more than 20 breeding sites and pooled areas, then taken to the insectarium of the Centre for Research in Infectious Diseases (CRID) in Yaoundé, where they were reared under standard laboratory conditions to the adult stage.

#### 2.2.2. Insecticide Bioassays and PBO-Based Synergist Test

Bioassays were conducted with female mosquitos obtained from field-collected larvae. Two- to five-day-old, unfed F_0_ females of *An. coluzzii* and *An. gambiae* were exposed for 1 h in WHO tube tests according to WHO procedures to 0.75% permethrin (1×). After observing resistance (mortality less than 95% [35]), we followed up with higher concentrations (3.5% (5×), then 7.5% (10×)) for the evaluation of resistance intensity [35].

To determine the possible implication of cytochromes P450 in the observed phenotypic resistance to pyrethroids, a test with synergist (PBO) that inhibits the effect of those P450s was done. F_0_ females were exposed to 4% PBO [36] for 1 h followed by an immediate second exposure to permethrin (0.75%) for another 1 h. The mortality was determined 24 h post-exposure and compared with mortality obtained for the mosquitos exposed to the 0.75% permethrin only using a Chi-square test of significance.

#### 2.2.3. Mosquito Selection

In the high pyrethroid resistance context of our collection areas, it was difficult to have enough susceptible samples from the field to run a comparative analysis between resistant and susceptible. Therefore, crossings between selected field permethrin-resistant and lab-susceptible strains from the same species were done as follows: *An. coluzzii* with Ngousso lab strain; *An. gambiae* with Kisumu lab strain. This follows the process described by Wondji et al. [37] and Cattel et al. [38]. To have distinguished phenotypes, F4 individual progenies obtained from those crosses were then segregated based on their resistance phenotype by exposing 3- to 5-day-old females to 0.75% permethrin at two different exposure times: 15 min and 90 min. After 24 h, dead individuals from the 15 min exposed batch were considered highly susceptible (HS) groups, while those who survived after the 90 min exposure time were considered highly resistant (HR) groups [37]. Samples were then stored on silica gel for the dead and in RNAlater^®^ for the survivors for subsequent molecular analysis.

### 2.3. Molecular Analysis

Genomic DNA (gDNA) was extracted using the LIVAK method [39] from 30 female mosquitos (F_0_) morphologically identified as belonging to the *An. gambiae* complex [40,41] originating from both collection sites and two sets each of 60 HR and HS mosquitos (F_4_) for both *An. coluzzii* and *An. gambiae*. Molecular identification was achieved through a polymerase chain reaction (PCR) described by Santolamazza et al. [42].

#### 2.3.1. Genotyping of *Kdr*-Mutation Alleles

To establish the frequency of each of the *kdr*-mutations (*L1014F*, *L1014S*, *N175Y*), genomic DNA of previously extracted samples were used to run the TaqMan assays described by Bass et al. [43] and Jones et al. [44]. The reaction mixture of 10 μL final volume containing 1 × Sensimix (Bioline), 1 × primer/probe mix and 1 μL template DNA was used for this assay. The probes were labeled with two distinct fluorophores [43]: FAM to detect the resistant alleles (*L1014F-kdr* and *L1014S-kdr*) and HEX to detect the susceptible allele (*L1014*). For *N1575Y*, the detection was performed by using the primer/probe described by Jones et al. [44]. The assay was performed following cycling conditions of 95 °C for 10 min, 40 cycles at 95 °C for 15 s and 60 °C for 1 min.

#### 2.3.2. Investigation of the Expression Profile of Six Main Candidate-Resistance Genes

The expression profiles of six genes were investigated by quantitative Reverse Transcriptase PCR (qPCR): *CYP6M2*, *CYP6P3*, *CYP9K1*, *CYP6Z1, CYP6Z2* and *CYP6P4*. For three biological replicates of 10 F_0_ females that recovered 24 h post-exposure to permethrin and 3 batches of 10 susceptible females of laboratory Ngousso (*An. coluzzii*) or Kisumu (*An. gambiae*), total RNA was extracted using the Arcturus PicoPure RNA isolation kit (Life Technologies, Carlsbad, CA, USA), followed by cDNA synthesis using Superscript III (Invitrogen^®^) with oligo-dT_20_ and RNase H, according to the manufacturer instructions. A qPCR reaction was performed following the protocol of Riveron et al. [45]. Fold changes and expression levels of each gene in field-caught permethrin-resistant (R) mosquitos and susceptible laboratory strain (S) samples were computed for both *An. coluzzii* and *An. gambiae* according to the 2-ΔΔCT method [46] following standardization with the housekeeping genes *ribosomal protein S7* (*RPS7*; VectorBase assession number: AGAP010592) and *Elongation factor* (VectorBase assession number: AGAP005128). The primers used here are those previously described by Mavridis et al. [47].

#### 2.3.3. Amplification and Direct Sequencing of Upstream and Full Gene Regions of CYP6M2 and CYP6P4

Exploiting the *An. gambiae* reference genome (VectorBase accession number: AGAP008212 and AGAP002867, respectively), both *An. coluzzii* and *An. gambiae* primers were designed for the amplification of a 1-kb upstream and full-gene length of *CYP6M2* and *CYP6P4* using Primer3 online software (v4.0.0; http://bioinfo.ut.ee/primer3/, accessed on 23 August 2019). The upstream region and full-gene length of each species were amplified from the gDNA of 10 samples of susceptible strain and two sets each of 10 HR and 10 HS female F_4_ mosquitos. The reaction mixture comprised 1.5 µL of 10× Buffer A (Kapa Biosystems, Wilmington, MA, USA); 0.75 µL of 25 mM MgCl_2_; 0.12 µL of 25 mM dNTPs; 0.12 µL of 5 U/µL Kapa Taq DNA polymerase; 0.51 µL each of the forward and reverse primer; 10.49 µL of double distilled water; and 1 µL of genomic DNA template. The thermocycling conditions of upstream and gene were the same for both *An. coluzzii* and *An. gambiae*. However, the annealing temperature (Ta) was different (Appendix A). Briefly, the conditions are: initial denaturation at 95 °C for 5 min followed by 35 cycles each of 30 s at 94 °C (denaturation), 2 min at primer annealing temperature, 1 min 30 s at 72 °C (elongation) and a final extension step at 72 °C for 10 min. All the PCR products *CYP6M2* and *CYP6P4* were purified using QIAquick^®^ PCR Purification Kit (Qiagen, Hilden, Germany) and ExoSAP-ITTM, respectively, according to the manufacturer’s protocol, and sent for sequencing at GENEWIZ UK LTD^®^ using the same PCR primers.

#### 2.3.4. In Silico Prediction of Promoter Region of CYP6M2 and CYP6P4

The *CYP6M2* and *CYP6P4* 1-kb upstream region was analyzed to identify any regulatory features, such as TATA box, CCAAT box and GC box, found in the sequence using GPMiner [48] as done to detect key resistance variants for CYP6P9a/b in *An. funestus* [28,29]. Additionally, the potential transcription factor binding sites (TFBSs) involved in the regulatory mechanism of resistance were identified in each upstream sequence [49,50], using a prediction software ALGGEN Promo (http://alggen.lsi.upc.es/cgi-bin/promo_v3/, accessed on 20 June 2022) in which positional weight matrices (PWM) are constructed from eukaryotic binding sites extracted from TRANSFAC [51], with a focus on human and mouse.

#### 2.3.5. Polymorphism-Based Analysis of Genomic DNA Sequence

Polymorphism analysis was carried out through the manual examination of the sequence traces using BioEdit version 7.2.3.0 [52] and/or nucleotides/amino acid differences from multiple sequence alignments. Genetic parameters, such as the number of haplotypes (h) and their diversity (Hd); number of polymorphic sites (S); and nucleotide diversity (π) were computed using DnaSP v6 [53]. Population Analysis with Reticulate Trees (PopART) version 1.7 software was used to construct haplotype networks showing the distribution of haplotypes per study site or species. MEGA 6.06 software was used to construct the Maximum Likelihood tree based on a specific parameter distance model according to the specific region with 1000 bootstrap replicates [54].

After polymorphism analysis, simple assays of Restriction fragment length polymorphism (RFLP) or Allele-specific PCR (AS–PCR) based on key mutations consistently found in higher frequencies in HR than in HS from F4 populations and the susceptible lab strain were designed to assess their frequency in field-caught populations.

#### 2.3.6. Detection of Key Polymorphisms in *An. gambiae* CYP6M2 and CYP6P4

PrimerQuest tools (https://eu.idtdna.com/Primerquest/ accessed on 20 June 2022) were used to design a PCR–RFLP and AS–PCR that could detect the key mutations found in *CYP6M2* and *CYP6P4* genes. These mutations were selected based on their frequency of occurrence by comparing HR, HS and susceptible lab strain Kisumu. AS–PCR assays were then designed for the genotyping of non-synonymous mutations A392S in *CYP6M2* and C168S in *CYP6P4*, using different sets of primers (Appendix A). Primers were designed manually to match the mutation, and an additional mismatched nucleotide was added in the 3rd nucleotide from the 3′ end of each inner primer to enhance the specificity as done previously by Tchouakui et al. [55]. RFLP PCRs, capable of discrimination between the mutant allele of the upstream region of *CYP6M2* and for both upstream and gene regions of *CYP6P4,* were designed based on specific restriction enzymes: BsrDI for the upstream region of *CYP6M2*, PvuII and EagI for *CYP6P4,* respectively. The restriction site was identified in the sequences of HR samples and absent in the sequences of Kisumu. Ten microliters of the digestion mix made of 1 µL of 10× NEBuffer 2.1, 0.2 µL of 2 U of respective restriction enzymes (New England Biolabs, Ipswich, MA, USA), 5 µL of PCR product and 3.8 µL of dH_2_O was incubated at specific temperatures for 2 h, as presented in Appendix A. According to the specific region, PCR products obtained in all the assays were separated on 2% agarose gel, stained with Midori Green Advance DNA Stain (Nippon Genetics Europe GmbH) and visualized using a gel imaging system to confirm the product sizes (Appendix A).

To evaluate the assay described above, the F4 progeny from a cross between field-caught permethrin-resistant from Nkolondom and susceptible laboratory strain was genotyped for 30–50 mosquitos and correlated with the established resistance phenotype using the odds ratio and Fisher’s exact test [56].

### 2.4. Data Analysis

The results of bioassays were interpreted based on percentage mortalities with standard error on the means (SEM) calculated and corrected when needed following WHO protocol [35]. Results of mortalities from synergist–permethrin exposure were compared with values obtained from exposure to pyrethroid alone using a two-tailed Chi-Square test of independence, with a level of significance set as *p* < 0.05, then implemented in GraphPad Prism 7.02 (GraphPad Inc., La Jolla, CA, USA). To investigate the association between specific mutations and the ability of the mosquitos to survive, Vassarstats was used to estimate the Odds Ratio (OR) based on a fisher exact probability test with a 2 × 2 contingency table.

## 3. Results

### 3.1. Species Identification and Susceptibility to Insecticides

The molecular identification of collected mosquitos confirmed the spatial distribution of the two species of *An. gambiae* complex: on 90 randomly tested samples per area, *An. coluzzii* was found exclusively in Ngousso, the urban area, whereas *An. gambiae* was found exclusively in Nkolondom, the peri-urban area.

F_0_ populations of *An. coluzzii* showed resistance to permethrin with a mortality rate of 8.33% ± 1.43% for all the samples tested. A pre-exposure to PBO revealed a 23.67% recovery of susceptibility (Figure 1a), with mortality rising to 32% ± 2.74% (χ^2^ = 64.37, *p* < 0.0001). The susceptibility of F_0_
*An. gambiae* to permethrin, showed a similar trend to what was observed in *An. coluzzii* with a mortality of 9.98% ± 3.43%. An increase in mortality with PBO was also observed, up to 25% ± 9.65% (χ^2^ = 56.39, *p* < 0.0001), representing a recovery of 16% (Figure 1a). These results suggest an involvement of cytochromes P450 in the resistance pattern observed in this population.

The resistance intensity test with 5× and 10× concentrations of permethrin revealed a high level of resistance in both species. In *An. coluzzii*, the mortality rate increased from 62.73% (±3.99) for 5× permethrin to 85.41% (±2.49) for 10× permethrin. In *An. gambiae*, the mortality rate increased from 75.02% (±3.49) to 96.33% (±9.79) for permethrin 5× and 10×, respectively (Figure 1a).

### 3.2. Expression Profile of CYP6M2 and CYP6P4

Resistant *Anopheles* populations from Nkolondom and Ngousso showed significant variation in metabolic gene expression between the two species for some of the tested genes. Both *CYP6P4* and *CYP6M2* were overexpressed in *An. coluzzii* F_0_ at a significant level (*p* < 0.05) with a fold change (FC) of 19.57 [95% CI 13.96–25.18] for *CYP6M2* and 6.73 [6.15–7.30] for *CYP6P4* compared to the susceptible laboratory colony (Figure 1b). In *An. gambiae*, those genes were also found to be overexpressed (*CYP6M2*: FC = 10.16 [7.86–12.46] and *CYP6P4*: FC = 23.62 [26.48–20.76]) (Figure 1b).

Moreover, in *An. coluzzii*, a significant over-expression of *CYP9K1* and *CYP6Z1* was observed in resistant compared to the Ngousso laboratory strain mosquitos (FC = 15.95 [9.48–22.43], *p* = 0.0168; FC = 11.53 [9.20–1386], *p* = 0.0015, respectively); and for *An. Gambiae, CYP6Z2* and *CYP6P3* were significantly overexpressed in resistant compared to Kisumu lab strain mosquitos (FC = 19.54 [14.83–24.24], *p* = 0.0024; FC = 2.23 [1.49–2.96], *p* = 0.0443, respectively) (Figure 1b).

### 3.3. Detection of the Knockdown Resistance Mutations

Among the three *kdr* mutations tested, only the *1014F kdr*-mutations were detected in any of the sites/species. This mutation was found fixed in the *An. gambiae* population from Nkolondom (1.00) and nearly fixed (0.93) in the *An. coluzzii* population from Ngousso (Appendix A). In F4 from crosses for both species, the HR samples showed the highest 1014F allelic frequency, 0.80 for *An. coluzzii* and 0.84 for *An. gambiae* (Figure 2), while in the HS sample, we obtained 0.07 for *An. coluzzii* to 0.10 for *An. gambiae* on 30 samples per batch.

### 3.4. Association of Polymorphisms in CYP6M2 and Pyrethroid Resistance

#### 3.4.1. Polymorphism Analyses of the Upstream Region of *CYP6M2*

To have a view of the potential regulatory elements driving the overexpression *of CYP6M2*, *An. gambiae* mosquito samples were used as a model. After cleaning and aligning the sequences, an indel of 8 bp (TAGTTACT) was found and is linked to the presence of the G/T mutation (Guanine replaced Thymine at position 316) in the HR (7/8 sequences) and HS samples (5/6). This Indel was absent for all the Kisumu tested as the G/T mutation is here absent. Appendix A shows the frequency of this indel across the groups used. The core promoter elements detected by GPMiner include TATA boxes (9 in the insertion carriers and 6 in the others), CCAAT box (1 in the insertion carriers and 2 in deletion carriers) and 1 GC box (Appendix A). Furthermore, the *CYP6M2* upstream region also contains transcription factors binding sites in all the samples tested, including 2 AhR/ARNT, 4 nrf2/MAF as well as several minority sites for GATA and MYB transcription factors (Appendix A).

The 932 bp upstream sequences of *CYP6M2* were aligned for 20 individuals of *An. coluzzii* (5 HR, 8 HS and 7 Ngousso strain) and 17 of *An. gambiae* (8 HR, 6 HS and 6 Kisumu lab strain) (Appendix A). For *An. coluzzii,* we observed that HR mosquito samples exhibited a low genetic diversity marked by a reduced haplotype diversity with one haplotype and no polymorphic sites detected (Table 1, Appendix A), compared to HS where 23 polymorphic sites were found in 4 haplotypes (h) with a high haplotype diversity (Hd = 0.60) and to the Ngousso strain (h = 3; Hd = 0.44). The haplotype network (Figure 3(a1)) and phylogenetic tree (Figure 3(a2)) showed a predominance of Hap_1 (75%) in the HR mosquitos compared to other groups with a marked difference in the haplotype diversity, suggesting that a possible selection could be acting on this gene or in nearby genes. Higher diversity is observed in the HS group with a high number of mutational steps between haplotypes, with the presence of two Indels of 3 bp and 8 bp. A positive Tajima’s D was observed in *An. Coluzzii*, indicating a decrease in population size and/or balancing selection.

In *An. gambiae*, no difference was observed when comparing the polymorphism parameters of the HR and HS groups, with a similar number of polymorphic sites (23 vs. 20) and a low haplotype diversity, 0.23 and 0.30 in HR and HS, respectively. However, a difference was observed between these groups and the susceptible lab Kisumu where the number of haplotypes is h = 4 with an equivalent high haplotype diversity Hd = 0.60 (Table 1). The haplotype network (Figure 3(b1)) and phylogenetic tree (Figure 3(b2)) showed that Hap_3 was the most represented (70%) and shared by HR and HS. Other haplotypes, such as Hap_1 (12%) and Hap_2 (6%), were found in the HS group, while Hap_5 was only found in the HR group (Figure 3(b1), Appendix A). This Hap_3 has the same sequence as the main haplotype (Hap_1) found in *An. coluzzii*. The negative values of the neutral test Tajima are indicative of purifying selection in HR and HS groups but are not statistically significant.

#### 3.4.2. Polymorphism Analyses of the Full-Gene Region of *CYP6M2*

The full-gene of *CYP6M2* was successfully sequenced in 21 and 28 individuals of both *An. coluzzii* (10 HR, 9 HS, 9 Ngousso strain) and *An. gambiae* (7 HR, 7 HS, 7 Kisumu lab strain), respectively (Appendix A). The nucleotide sequence analysis of this fragment exhibited a lower number of haplotypes (8) and polymorphic sites (20) in *An. coluzzii* HR as compared to 14 haplotypes and 27 polymorphic sites in HS individuals (Table 2, Appendix A). This is evident in the lower number of synonymous (11 and 13) and non-synonymous (7 and 10, respectively) mutations but not when compared with lab strain Ngousso (Table 2). The haplotype network (Figure 4(a1)) and phylogenetic tree (Figure 4(a2)) analysis showed a predominance of Hap_3 (21%), which was shared by all mosquito groups tested. Most haplotypes are separated from each other by 1 to 4 mutational steps and have probably derived from common ancestral susceptible strains. In *An. gambiae*, HR mosquitos showed 36 polymorphic sites with 14 non-synonymous mutations compared to 44 polymorphic sites giving all 14 non-synonymous mutations in HS (Table 2). The haplotype network and phylogenetic profile showed two clear clusters among 20 *An. gambiae* haplotypes, a resistant cluster with 6 haplotypes and a susceptible cluster with 5 haplotypes (Figure 4(b1,b2)). But, in both species, a star-like shape of the haplotype network indicates a high frequency of intermediate variants, which can be an indication of population expansion.

Moreover, the GCC-> TCC polymorphism in codon *392-CYP6M2*, inducing an amino acid change of alanine to serine A392S (Appendix A), was observed in *An. gambiae* samples, with a frequency of 71.43% (5/7) in the HR and 57.14% (4/7) in HS but not observed in Kisumu lab strain (0/7) as shown in Appendix A. Samples carrying the *392A* allele were more polymorphic (S = 33, Hd = 0.89 and π = 0.0089; Figure 4(b3)) than *392S* allele carriers (S = 25, Hd = 0.88 and π = 0.0059; Figure 4(b3)), suggesting a purifying or balancing selection in mosquitos carrying the A^392^S and confirmed by a positive value obtained for the Tajima’s D and Fu and Li statistics (Appendix A).

### 3.5. Genetic Polymorphisms Analysis of CYP6P4 Associated to Pyrethroid Resistance

#### 3.5.1. Polymorphism Analyses of the Upstream Region of *CYP6P4*

The *CYP6P4* upstream region (the intergenic region between *CYP6P4* and *CYP6P5*) was amplified and sequenced for individual HR and HS mosquitos. As above, *An. gambiae* mosquito samples were used as a model to characterize this intergenic region. After aligning sequences, an indel of 7 bp (GGGGTGC) was consistently observed with a deletion associated with an A/T substitution in all the HR (6/6) and less than 50% in HS, and present in susceptible strain Kisumu. Appendix A shows the frequency of this indel across the groups used. Overall, the core elements detected by GPMiner include TATA boxes (5), CCAAT box (3), and 3 GC box (Appendix A). The *CYP6P4* upstream also exhibits several transcription factors binding sites in *An. gambiae* (Appendix A), including several sites for GATA, MYB or AHR and nrf2/MAF (5 in HR samples and 4 in the other groups). This additional nrf2/MAF site detected in all the HR samples included an A/T substitution at position 273 (Appendix A).

Regarding the 870-bp of the intergenic region between *CYP6P4* and *CYP6P5*, a comparative analysis was done using 18 sequences (6 HR, 4 HS and 8 Ngousso) and 22 (6 HR, 9 HS and 7 Kisumu), respectively, for *An. coluzzii* and *An. gambiae* (Appendix A). In *An. coluzzii*, HR and HS mosquitos revealed a high haplotype diversity of 0.88 and 0.71, respectively (Table 3). The haplotype 2 was predominant (39%) and shared by all groups, while Hap_3 (22%) and Hap_1 (5%) were specific to susceptible Ngousso lab strain; Hap_4 (5%) and Hap_5 (5%) to HS; and the rest to HR (Figure 5(a1,a2), Appendix A). In addition, most HR haplotypes are separated from each other and from other haplotypes by a large number of mutational steps confirming the high genetic diversity observed in their sequences.

In *An. gambiae*, the number of haplotypes for this region was low in HR (2 haplotypes) with a low Hd of 0.485 for 2 polymorphic sites, compared to HS (7 haplotypes) with Hd = 0.85 (Table 3, Appendix A) and 17 polymorphic sites, suggesting a possible ongoing directional selection in resistant mosquitos. The haplotype Hap_2 (34%) was shared by all groups with Hap_3 (13%), present only in the susceptible groups (HS and susceptible lab strain Kisumu), while Hap_11 (9%) was only found in the HR groups (Figure 5(b1,b2)). This Hap_2 corresponds to the main haplotype (Hap_2) found in *An. coluzzii*. This reduced diversity in *An. gambiae* HR groups supports the hypothesis of ongoing selection, but it is not confirmed by the positive values of both neutrality tests. In addition, based on the allelic profile at position 273 (Appendix A), the genetic variability parameters showed that a higher number of haplotypes occurs in *273-A* allele carriers (h = 7), with an equivalent high haplotype diversity, Hd = 0.84, and nucleotide diversity, π = 0.0096, while the lowest haplotype diversity and nucleotide diversity (h = 5, Hd = 0.58, π = 0.0013) were found in *273*-*T* allele carriers (Figure 5(b3), Appendix A).

#### 3.5.2. Polymorphism Analyses of the Full-Gene Region of *CYP6P4*

Analysis of variability of a 1051 bp-fragment of the *CYP6P4*-gene (1521 bp) was done on 58 individual sequences from *An. coluzzii* (9 HR, 10 HS, 10 Ngousso strain) and *An. gambiae* (10 HR, 9 HS, 10 Kisumu lab strain) (Appendix A). In *An. coluzzii*, HR mosquitos showed a lower number of haplotypes (8) than the HS individuals (9) (Table 4, Appendix A). The haplotype network analysis and phylogenetic tree for this fragment showed that the observed haplotypes are genetically diverse and do not cluster according to resistance phenotype (Figure 6(a1,a2)). However, Hap_2 was predominant (14%) and shared by all sample groups tested, the others being distributed as shown in Figure 6(a1,a2) below. The absence of predominant haplotypes indicates that this gene is not subject to selection pressure in *An. coluzzii* specimens from Yaoundé, Cameroon.

The HR sequences of *An. gambiae* exhibited 11 polymorphic sites with 12 haplotypes (Table 4, Appendix A). For the HS and Kisumu lab strain, sequences were most polymorphic with 13 and 16 polymorphic sites, respectively (Appendix A). The maximum likelihood phylogenetic tree generated shows a small cluster associated with HR mosquitos (4 out of 10 sequences) (Figure 6(b1,b2)). Hap_24 was predominant (14%) followed by Hap_14 (5.2%) and Hap_19 (3.4%), shared by the HR and HS samples. On the other hand, Hap_4 (5.2%) and Hap_10 (5.2%) were predominant in HS specimens or the Kisumu specimens.

In addition, two synonymous and one non-synonymous mutations were identified (Appendix A) in the *An. gambiae CYP6P4* gene. The synonymous substitution GCG- > GCA in codon 115 was linked to the mutation of TGC- > AGC in codon 168, which led to an amino acid change of cysteine to serine (*C168S*) substitution; the two mutations were found with a frequency of 70% in the HR, 28% in HS and 15% in Kisumu (Appendix A). The carriers of *168C* allele show the highest polymorphism with 23 haplotypes (Hd = 0.97, π = 0.0005) in 17 polymorphic sites (Appendix A) compared to samples carrying the *168S* allele where 11 haplotypes (Hd = 0.86, π = 0.0003) (Figure 6(b3) and Appendix A). The non-synonymous GGC- > GGT at codon 144 was found with a frequency of 95% HR compared to HS and Kisumu of 50% and 20% frequency respectively (Appendix A).

### 3.6. Distribution of Key Mutations Found in CYP6M2 and CYP6P4 Genes among the Populations of An. gambiae

The presence of mutations in the upstream and the full-gene length regions of *CYP6M2* and *CYP6P4* were in the HS and HR strains, and direct field-collected samples from Nkolondom (Table 5, Appendix A) were determined. No significant link between any of the mutations was identified, and pyrethroid resistance was detected (Table 5).

## 4. Discussion

Insecticide resistance is a major threat to the effectiveness of insecticide-based malaria control tools. Besides the already well-studied target site mutations, metabolic resistance is one of the most important mechanisms. Although the expression of cytochrome P450 genes, such as *CYP6M2* and *CYP6P4,* capable of breaking down the insecticide before it reaches its target is a well-known mechanism of insecticide resistance, the underlying factors modulating the expression of this enzyme family are poorly understood. The present study analyzed the polymorphisms of these two cytochrome-P450s to detect potential mutations that are linked to their overexpression and may be used as resistance markers in two main malaria vectors populations in Yaoundé, Cameroon.

The study shows resistance to permethrin in both species in line with previous reports in different studies in the country [23,24,25,57,58] and highlighted an escalation of resistance over years in the country with observed resistance to higher insecticide concentrations [32]. Additionally, pre-exposure to PBO reveals a partial recovery of susceptibility to permethrin in both collection sites, indicating an involvement of oxidases, such as cytochromes P450 in the observed resistance. However, the recovery of susceptibility after PBO is moderate (<20%) showing that other mechanisms than P450s are involved, including kdr, which were found fixed in *An. gambiae*. The escalation in resistance could be explained by the insecticide selection pressure due to the scale up of control interventions with the distribution of ITNs taking place in Cameroon since 2011 and also by the presence of pesticide residues or components associated with anthropogenic activities (soapy water, organic waste) in the urban breeding sites. Those residues may activate certain detoxication enzymes in mosquitos by oxidative stress, enzymes being also active against insecticides molecules [59,60].

The consistent over-expression of *CYP6M2* and *CYP6P4* genes in field permethrin-resistant *An. gambiae* and *An. coluzzii*, among all the up-regulated genes found in this study, shows their link with resistance patterns in Cameroon. These two genes have been firstly recorded upregulated in Cameroon DDT and pyrethroid resistant *An. gambiae* and *An. coluzzii* mosquitos [22], and more recently, in *An. gambiae* populations from the same peri-urban area [23,24]. Meanwhile, Ibrahim et al. [14] have already established that *CYP6P4* is responsible for resistance to permethrin in *An. arabiensis* populations from Central Africa. Many studies have shown that the up-regulation of certain P450s can be modulated by mutations of cis-regulatory elements in the promoter regions of P450 genes and/or changes in the expression level of transcription factors binding to these cis-regulatory elements [28,61,62]. Knowing that other mechanisms, such as allelic variations of resistance genes, may also be involved, we decided in this study to amplify around 1 kb of the putative promoter and full-gene region of *CYP6M2* and *CYP6P4* in both *An. coluzzii* and *An. gambiae* to detect any variations that might influence their expression in resistant individuals.

Polymorphism analysis of *CYP6M2* sequences in *An. coluzzii* from Ngousso, Yaoundé, revealed that for both the upstream and the full-gene length regions, the HS strain was more polymorphic than the HR, but the predominant haplotype was shared by both HR and HS. In *An. gambiae* the upstream region of *CYP6M2* in HS mosquitos showed a similar polymorphism to that in HR, and this, coupled with the high diversity in HR samples compared to HS, suggests that there is not a strong selective sweep acting upon *CYP6M2* in either species in Yaoundé. The absence of polymorphisms associated with resistance in the coding region or in the 1 kb upstream may indicate that the emergence of *CYP6M2*-associated resistance is due to selection pressures acting on genes encoding distantly related regulatory proteins in *trans* position, as concluded in a similar study comparing *CYP6M2* sequences of *An. gambiae* s.l. sampled from 13 countries [18]. This larger study identified 6 amino acid polymorphisms within *CYP6M2*, none of which were detected in the current study. The mutation A392S was found to be the most frequent in F_4_ HR (50%) of *An. gambiae* from Nkolondom crossing and was also detected in field F_0_ populations (7%).

A detailed analysis of the polymorphisms in both the upstream and full-gene regions of *CYP6P4* revealed that the HS was more polymorphic than HR samples of *An. gambiae* populations. The key amino acid change (C168S) observed in *CYP6P4* and substitution mutations in the nrf2/MAF TFBs of their upstream region appear to be subject to positive selection in *An. gambiae* populations and could signal a directional selection as seen for *CYP6P9a/b* [45].

In the full-gene region, one major amino-acid change at codon 168 where cysteine (C) is replaced by serine (S) were found in almost all the HR sequences. This amino-acid change was different from what has been seen in Uganda with the I^236^M mutation [20], with only the susceptible allele 236I found in all the *An. coluzzii* and *An. gambiae* sequence from Yaoundé, Cameroon. The impact of such allelic variation on the metabolic efficiency of detoxication genes has previously been demonstrated *in An. funestus* for P450s, such as *CYP6P9a/b* [27], and is similar to the case of *CYP6A2* in *Drosophila melanogaster* for which 3 amino acid substitutions located close to the active site in the allele predominant in DDT-resistant flies have been shown to confer the increased metabolism of DDT [62]. Further studies on the variants are needed to determine whether they vary in their ability to detoxify pyrethroids.

## 5. Conclusions

This study explored the genetic diversity of two cytochrome P450s associated with pyrethroid resistance in *An. gambiae* s.l. revealing some potentially interesting molecular markers in *CYP6P4* that merit further investigation in the resistance context in Cameroon. No molecular markers associated with resistance were found in *CYP6M2* in either species agreeing with previous studies. Further studies need to be performed to detect the specific genetic variants driving the overexpression of both *CYP6M2* and *CYP6P4* in the central African population of *An. gambiae* s.l., exploring structural variants and both cis- and trans- regulatory loci. Such work will help design robust DNA-based assays to detect and track resistance.

## Figures and Tables

**Figure 1 genes-14-00052-f001:**
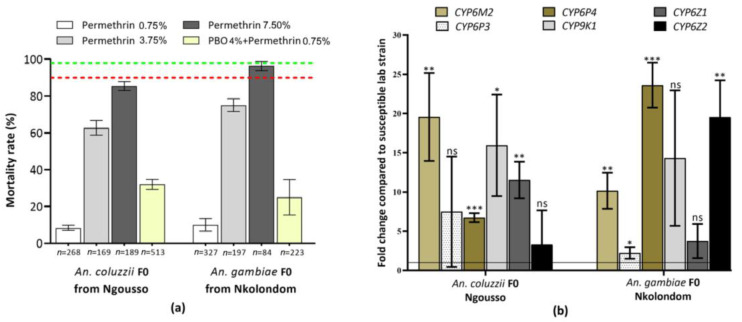
Resistance profile of *An. coluzzii* and *An. gambiae* to Permethrin. (**a**) Activities of PBO combined to permethrin on *An. coluzzii* from Ngousso and *An. gambiae* from Nkolondom, Cameroon. Data are shown as median ± SEM. (**b**) Differential expression of some metabolic resistance genes among the Ngousso and Nkolondom resistant mosquito populations as compared with Ngousso or Kisumu laboratory strain, respectively, to the species. Degree of significance of differential expression in comparison with susceptible strains: “***” < 0.0005 *p*-values, “**” < 0.005, “*” < 0.05, “ns”: non-significant.

**Figure 2 genes-14-00052-f002:**
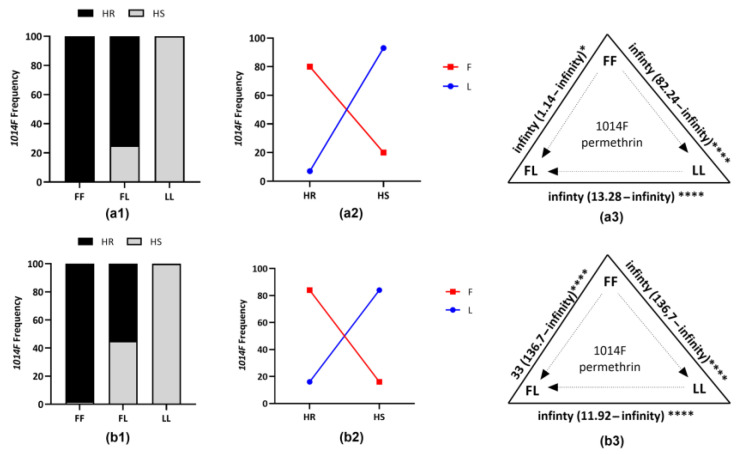
Correlation between L1014F kdr and permethrin-resistance phenotype in *An. coluzzii* and *An. gambiae* from F4. (**a1**,**b1**) Distribution of kdr L1014F genotypes in dead and alive F4 mosquitos from *An. coluzzii* and *An. gambiae* crossings, respectively, after WHO bioassays with 0.75% permethrin. (**a2**,**a3**,**b2**,**b3**) Association between frequency of 1014F and ability of F4 mosquitos, respectively, from *An. coluzzii* and *An. gambiae* crossings to survive to WHO bioassays with 0.75% permethrin, showing a very strong link between 1014F kdr mutation and permethrin-resistance phenotype. HR, Highly resistant; HS, Highly susceptible; FF, homozygous resistant; FL, heterozygous resistant; LL, homozygous susceptible; F, resistant allele (L1014F); L, susceptible allele (L1014). Significance is shown by * *p* < 0.05, **** *p* < 0.0001, as estimated using Fisher’s exact test.

**Figure 3 genes-14-00052-f003:**
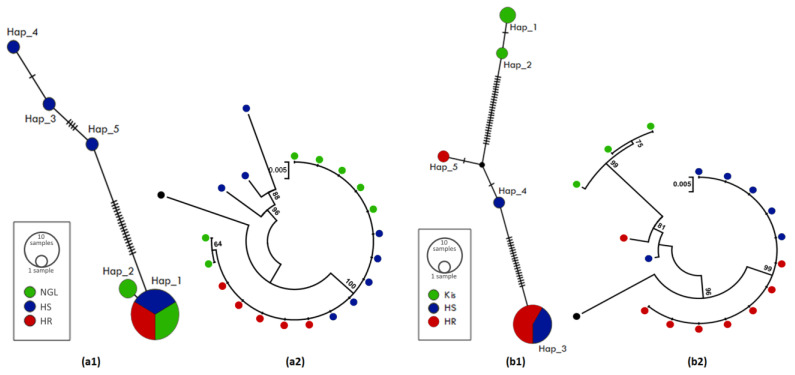
Patterns of the genetic variability and polymorphism of the upstream region of *CYP6M2*. (**a1**) TCS-network of the *An. coluzzii* sequences showing haplotypes and their respective frequencies (Hap_1, inset); (**a2**) a maximum likelihood phylogenetic tree of *An. coluzzii* sequences generated using Tamura 3-parameter model; (**b1**) TCS-network of the *An. gambiae* sequences showing haplotypes and their respective frequencies (Hap_3, inset); (**b2**) a maximum likelihood phylogenetic tree of *An. gambiae* sequences generated using Tamura 3-parameter model. Hap, Haplotype; NGL, Ngousso lab strain; HR, Highly resistant; HS, Highly susceptible; Kis, Kisumu lab strain.

**Figure 4 genes-14-00052-f004:**
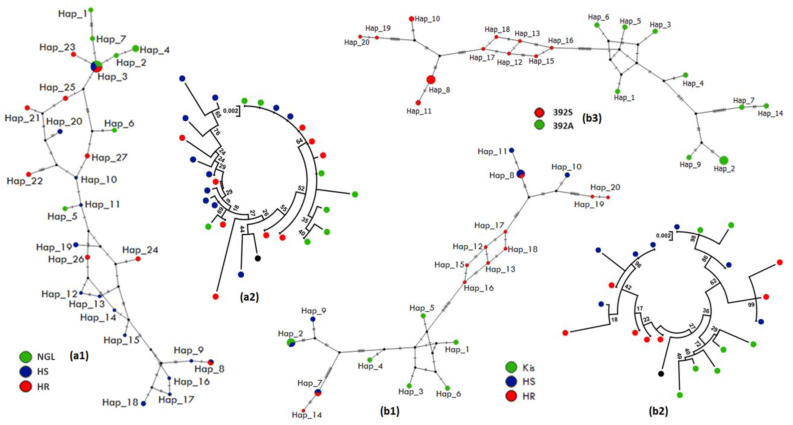
Pattern of the genetic variability and polymorphism of full-gene length of *CYP6M2*. (**a1**) TCS-network of the *An. coluzzii* sequences showing haplotypes and their respective frequencies (Hap_3, inset); (**a2**) a maximum likelihood phylogenetic tree of *An. coluzzii* sequences generated using Tamura 3-parameter model; (**b1**) TCS-network of the *An. gambiae* sequences showing haplotypes and their respective frequencies (Hap_7 and Hap_8, inset); (**b2**) a maximum likelihood phylogenetic tree of *An. gambiae* sequences generated using Kimura 2-parameter model; (**b3**) TCS-network of the A392S-*CYP6M2* mutation per allele in *An. gambiae*. Hap, Haplotype; NGL, Ngousso lab strain; HR, Highly resistant; HS, Highly susceptible; Kis, Kisumu lab strain.

**Figure 5 genes-14-00052-f005:**
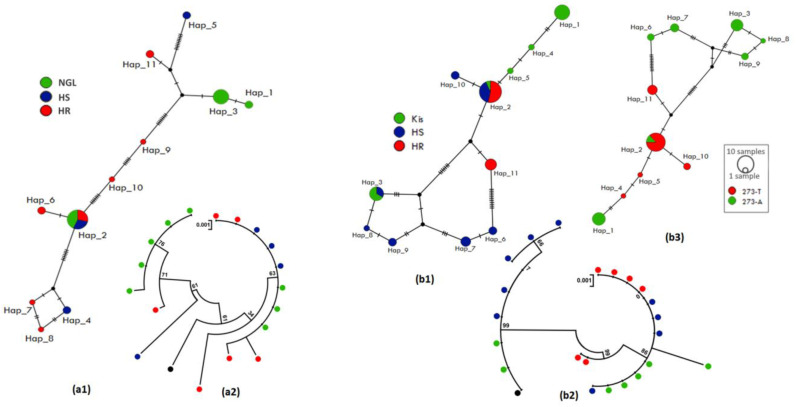
Pattern of the genetic variability and polymorphism of the upstream region of *CYP6P4*. (**a1**) TCS-network of the *An. coluzzii* sequences showing haplotypes and their respective frequencies (Hap_2, inset); (**a2**) a maximum likelihood phylogenetic tree of *An. coluzzii* sequences generated using Kimura 2-parameter model; (**b1**) TCS-network of the *An. coluzzii* sequences showing haplotypes and their respective frequencies for *An. gambiae* (Hap_2, inset); (**b2**) a maximum likelihood phylogenetic tree of *An. gambiae* sequences generated using Juke- Cantor model. (**b3**) TCS-network of the upstream region of *CYP6P4* mutation per allele in *An. gambiae*. Hap, Haplotype; NGL, Ngousso lab strain; HR, Highly resistant; HS, Highly susceptible; Kis, Kisumu lab strain.

**Figure 6 genes-14-00052-f006:**
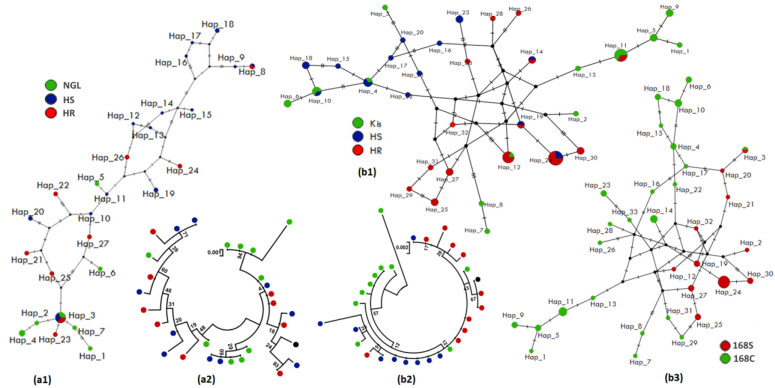
Pattern of the genetic variability and polymorphism of full-gene length of *CYP6P4*. (**a1**) TCS-network of the *An. coluzzii* sequences showing haplotypes and their respective frequencies (Hap_2, inset); (**a2**) a maximum likelihood phylogenetic tree of *An. coluzzii* generated using Tumura 3-parameter model; (**b1**) TCS-network of the *An. gambiae* sequences showing haplotypes and their respective frequencies (Hap_24, inset); (**b2**) a maximum likelihood phylogenetic tree of *An. gambiae* sequences generated using Tumura 3-parameter model; (**b3**) TCS-network of the C168S-CYP6P4 mutation per allele in *An. gambiae*. Hap, Haplotype; NGL, Ngousso lab strain; HR, Highly resistant; HS, Highly susceptible; Kis, Kisumu lab strain.

**Table 1 genes-14-00052-t001:** Summary statistics for the polymorphism of *CYP6M2* upstream region in permethrin highly susceptible and highly resistant.

Species (Size)	Populations	2n	S	H (Hd)	Π	D	D *
*An. coluzzii* (932 bp)	HR F_4_	10	0	1 (0.000)	0.00000	n.a	n.a
	HS F_4_	16	23	4 (0.600)	0.01221	2.387 *	1.592 **
	Ngousso	14	1	2 (0.440)	0.00049	0.842	1.581
	All	40	24	5 (0.431)	0.00645	0.103	1.722 **
*An. gambiae* (932 bp)	HR F_4_	16	23	2 (0.233)	0.00583	−0.917	1.623 **
	HS F_4_	12	20	2 (0.303)	0.00659	−0.373	1.593 **
	Kisumu	6	1	2 (0.533)	0.00057	0.851	1.053
	All	34	49	5 (0.492)	0.01831	1.474	1.859 **

2n, number of unphased sequences; D, Tajima’s statistics; D *, Fu and Li’s statistics; H, number of haplotypes; π, nucleotide diversity; S, number of polymorphic sites; n.a = non-attributed; Hd, haplotype diversity; * = *p* < 0.05; ** = *p* < 0.02.

**Table 2 genes-14-00052-t002:** Summary statistics for full *CYP6M2*-gene polymorphism in permethrin highly susceptible and highly resistant.

Species (Size)	Populations	2n	S	H (Hd)	Syn	NSyn	π	D	D *	pKa/pKs
*An. coluzzii* (1104 bp)	HR F_4_	18	20	8 (0.915)	11	7	0.0064	0.799	1.497 **	0.46
	HS F_4_	20	27	14 (0.953)	13	10	0.0099	1.715	1.592 **	0.38
	Ngousso	18	17	7 (0.889)	9	7	0.0045	0.059	1.4973 **	0.37
	All	56	32	27 (0.944)	15	13	0.0097	1.238	1.8745 **	0.34
*An. gambiae* (1528 bp)	HR F_4_	14	36	11 (0.967)	17	14	0.0102	1.621	1.493 **	0.26
	HS F_4_	14	44	6 (0.879)	23	14	0.0122	1.515	1.493 **	0.19
	Kisumu	14	22	6 (0.879)	14	5	0.0064	1.728	1.493 **	0.11
	All	42	51	20 (0.948)	26	23	0.0124	2.124 *	1.929 **	0.24

2n, number of unphased sequences; D, Tajima’s statistics; D *, Fu and Li’s statistics; H, number of haplotypes; n.a = non-attributed; Hd, haplotype diversity; Syn, Synonymous mutations; Nsyn, Non-synonymous mutations; π, nucleotide diversity; S, number of polymorphic sites; pKa, Synonymous polymorphism per site; pKs, non-Synonymous polymorphism per site; * = *p* < 0.05; ** = *p* < 0.02.

**Table 3 genes-14-00052-t003:** Summary statistics for polymorphism of *CYP6P4* upstream region in permethrin highly susceptible and highly resistant.

Species (Size)	Population	2n	S	H (Hd)	π	D	D *
*An. coluzzii* (870 bp)	HR F_4_	12	22	7 (0.879)	0.00935	0.3557	1.509
	HS F_4_	8	20	3 (0.714)	0.01071	1.1576	1.558 **
	Ngousso	16	15	3 (0.630)	0.00822	2.3509 *	1.509 **
	All	36	29	11 (0.803)	0.01015	0.8198	1.749 *
*An. gambiae* (880 bp)	HR F_4_	12	2	2 (0.489)	0.00112	1.356	1.195
	HS F_4_	18	17	7 (0.856)	0.00952	2.529 **	1.348 **
	Kisumu	14	15	5 (0.703)	0.00754	1.537	1.288 **
	All	44	19	11 (0.838)	0.00828	2.047 *	1.655 **

2n, number of unphased sequences; D, Tajima’s statistics; D *, Fu and Li’s statistics; H, number of haplotypes; π, nucleotide diversity; S, number of polymorphic sites; n.a. = non-attributed; Hd, haplotype diversity; * = *p* < 0.05; ** = *p* < 0.02.

**Table 4 genes-14-00052-t004:** Summary statistics for polymorphism of *CYP6P4*-gene in permethrin highly susceptible and highly resistant.

Species (Size)	Population	2n	S	H (Hd)	Syn	NSyn	π	D	D *	pKa/pKs
*An. coluzzii* (1037 bp)	HR F_4_	18	21	8 (0.915)	19	2	0.0076	1.114	1.246 **	0.002
	HS F_4_	20	20	9 (0.921)	21	0	0.0078	1.365	1.268	0.000
	Ngousso	20	25	9 (0.922)	19	7	0.0076	0.095	1.246 *	0.078
	All	58	33	25 (0.963)	25	9	0.0084	0.610	1.614 **	0.003
*An. gambiae* (1051 bp)	HR F_4_	20	11	12 (0.905)	10	1	0.0041	1.324	1.007	0.030
	HS F_4_	18	13	13 (0.967)	11	2	0.0048	1.275	0.667 *	0.055
	Kisumu	20	16	13 (0.947)	13	3	0.,0059	1.415	1.007 **	0.070
	All	58	17	33 (0.969)	14	3	0.0062	2.385 *	1.632 **	0.065

2n, number of unphased sequences; D, Tajima’s statistics; D *, Fu and Li’s statistics; H, number of haplotypes; n.a = non-attributed; Hd, haplotype diversity; Syn, Synonymous mutations; Nsyn, Non-synonymous mutations; π, nucleotide diversity; S, number of polymorphic sites; pKa: Synonymous polymorphism per site; pKs: non-Synonymous polymorphism per site; * = *p* < 0.05; ** = *p* < 0.02.

**Table 5 genes-14-00052-t005:** Frequencies of key mutations found in *CYP6M2* and *CYP6P4* in *An. gambiae*.

**Groups**	**Indel of 8 bp in Upstream of *CYP6M2***			* **N** *	* **2N** *	**Allele**		**OR (CI)**	***p* Value **
	**D/D**	**D/I**	**I/I**			**D**	**I**		
Kisumu	0	3	14	17	34	0.09	0.91	11.40 (5.18–25.11)	<0.0001
F_0_	6	8	5	19	38	0.53	0.47		
HR F_4_	18	21	15	54	108	0.53	0.47	0.73 (0.42–1.27)	0.32
HS F_4_	10	32	16	58	116	0.45	0.55		
**Groups**	**Amino Acid Residue at 392 *CYP6M2*-Gene**			** *N* **	** *2N* **	**Allele**		**OR (CI)**	***p* Value**
	**S/S**	**S/A**	**A/A**			**S**	**A**		
Kisumu	0	8	22	30	60	0.13	0.87	0.50 (0.19–1.32)	0.23
F_0_	1	6	49	56	112	0.07	0.93		
HR F_4_	13	23	14	50	100	0.49	0.51	0.59 (0.33–1.03)	0.09
HS F_4_	8	21	22	51	102	0.36	0.64		
**Groups**	**Indel of 7 bp in Upstream of *CYP6P4***			** *N* **	** *2N* **	**Allele**		**OR (CI)**	***p* Value**
	**t/t**	**t/a**	**a/a**			**t**	**a**		
Kisumu	0	6	6	12	24	0.25	0.75	1.46 (0.76–2.80)	0.32
F_0_	3	5	11	20	40	0.29	0.71		
HR F_4_	5	25	16	46	92	0.38	0.62	1.39 (0.79–2.44)	0.32
HS F_4_	8	21	11	40	80	0.46	0.54		
**Groups**	**t/c Mutation at Codon 144 *CYP6P4*-Gene**			** *N* **	** *2N* **	**Allele**		**OR (CI)**	***p* Value**
	**t/t**	**t/c**	**c/c**			**t**	**c**		
Kisumu	0	8	16	24	48	0.21	0.79	1.46 (0.76–2.80)	0.32
F_0_	3	5	12	20	40	0.28	0.72		
HR F_4_	25	19	0	44	88	0.78	0.22	0.52 (0.28–0.98)	0.06
HS F_4_	16	23	3	42	84	0.65	0.35		
**Groups**	**Amino Acid Residue at 168 *CYP6P4*-Gene**			** *N* **	** *2N* **	**Allele**		**OR (CI)**	***p* Value**
	**C/C**	**C/S**	**S/S**			**C**	**S**		
Kisumu	0	8	12	20	40	0.20	0.80	1.00 (0.50–1.99)	1.00
F_0_	0	10	15	25	50	0.20	0.80		
HR F_4_	3	22	5	30	60	0.47	0.53	1.00 (0.57–1.74)	1.00
HS F_4_	6	16	8	30	60	0.47	0.53		

D, Deletion; I, insertion; S, Serine; A, Alanine; C, Cysteine; S, Serine; t, thymine; a, adenine; c, cytosine; N = total number of samples tested; OR, odds ratio; CI, confidence interval.

## Data Availability

All data generated or analyzed during this study are included within the article and its additional files. The sequences generated have been deposited in the GenBank database (study accession numbers: OP779727-OP779746; OP797835-OP797861; OP797862-OP797886; OP811275-OP811307; OP846110-OP846114; OP846115-OP846119; OP857148-OP857158; OP857159-OP857169).

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
