# Peer review of "Genetic Diversity of Cytochrome P450s *CYP6M2* and *CYP6P4* Associated with Pyrethroid Resistance in the Major Malaria Vectors *Anopheles coluzzii* and *Anopheles gambiae* from Yaoundé, Cameroon"

_genes, 2022, doi:10.3390/genes14010052_

Round 1

Reviewer 1 Report

The present manuscript by Togeum et al. describes the genetic diversity of Cytochrome P450s associated with pyrethroid resistance in  Anopheles coluzzii and Anopheles gambiae. They found that CYP6P4-associated resistance is potentially driven by changes in the upstream region. Although the authors have successfully explained the possible reason for resistance and performed enough experiments to support their study, I have a few concerns that need to be addressed before the acceptance of this manuscript. 

Line 122: Could you please elaborate on the method of segregation?

Line 124: Why were these specific concentrations selected as low to higher concentrations? did the authors perform a pilot study before the experiments?

Line 127: Please explain the reason for selecting 4% PBO

Line 137-143: How it happened? It seems something fishy here. Kindly check the statement again. If 0.75% permethrin kills 20% F4 individuals at 15 min exposure, then it should be resistant rather than susceptible.

Line 177:  Authors selected the RPS7 gene as an internal control; however, it has not been mentioned in MS whether it was already validated by other studies or in the present study. If it was validated by other studies, then please cite and add references. Moreover, according to standard criteria, at least 2-3 internal controls should be used to get reliable results, as using only genes might provide misleading results

Author Response

Line 122: Could you please elaborate on the method of segregation? Done

Line 124: Why were these specific concentrations selected as low to higher concentrations? did the authors perform a pilot study before the experiments? The concentrations used to evaluate insecticide resistance intensity are defined by WHO (5x and 10x time the standard 1x concentration)

Line 127: Please explain the reason for selecting 4% PBO A short sentence was added, but it is well explained in lines 54-56.

Line 137-143: How it happened? It seems something fishy here. Kindly check the statement again. If 0.75% permethrin kills 20% F4 individuals at 15 min exposure, then it should be resistant rather than susceptible. The individuals who died after only 15min exposure are considered highly susceptible, while those surviving after 90min exposure to insecticide are considered highly resistant. Those criteria are for the segregation of both lines to be used for further analysis according to previous work (Wondji et al 2007)

Line 177:  Authors selected the RPS7 gene as an internal control; however, it has not been mentioned in MS whether it was already validated by other studies or in the present study. If it was validated by other studies, then please cite and add references. Moreover, according to standard criteria, at least 2-3 internal controls should be used to get reliable results, as using only genes might provide misleading results. Good point. The references were added for the internal controls used. There are two in our study: RPS7 and Elongation factor.

Reviewer 2 Report

Fotso-Toguem et al present a study attempting to dissect the insecticide resistance mechanisms in populations of An. gambiae and An. coluzzi in Cameroon through a combination of crossings, genotyping and qPCR strategies. Their focus is on two cytochrome P450 genes previously found to be upregulated in many resistant populations and they attempt to look for the molecular drivers of over-expression in these genes.

These studies are always complicated by the lack of susceptible field caught mosquitoes so the authors do what the majority of previous studies have been forced to do and use susceptible colonies for comparators in qPCR or, in this case, crossing. It is somewhat of a shame that this study went only to F4 in the crossing strategy – it does seem to me that further generations would have been advisable particularly for the association tests that they are conducting. I do think that they over-interpret some of their results (or at least over-discuss when there is clearly no evidence for any difference between the HS and HR groups) however, they do then adequately reel this in for the discussion when talking about that this indicates trans acting or distant cis acting factors may be involved. I do actually query whether the important cis elements for cyp6m2 and cyp6p4 are actually immediately 5’ or whether this cluster might have regulatory regions upstream of the whole cluster. Perhaps the authors could consider this in their discussion.

On the whole the manuscript is very well written although there are some areas that could be improved to aid the reader.

Whilst I don’t have any really major issues the following things would, I think, aid the reader if they were clarified or if additions were made, and then there are minor issues below:

Mosquito collection (L110 onwards). How was sampling of families avoided?

L207. ‘known binding sites’. In what species? This is important for accurately detecting such sequences.

L215. Why is 0.5 taken as low? On what basis?

L250. I am unclear what is meant by continuous variables here since mortality in WHO bioassays is binary – dead or alive. So what is the continuous variable?

L273-277 talks about tests with 5x and 10x concentrations of insecticide but this is not mentioned in the methods. Please do so. Rather than saying 5x and 10x give actual concentration. If 0.75% is standard then presumably 3.75% and 7.5%

Figure 2. Using R and S for alleles is extremely confusing in this context since you use R and S also for phenotype. For alleles, use Ser and Phe (or S and F) to avoid confusion. Ensure that this is done throughout the ms. Figure 2 also talks about hybrids but there is no information in the methods that any hybrids were generated. An. gambiae was always crossed with gambiae and coluzzi with coluzzi. This needs attending to or explaining better.

Section 3.4.1 would benefit from a figure to depict where the polymorphisms being mentioned are with respect to the TSS. All polymorphisms should be described with rs numbers (where available) or genome positions where not.

Figure 3. These colours need changing. They are difficult to discriminate – particularly for people with colour blindness. Somewhere in the text it would be useful to mention whether any An. coluzzi haplotypes are shared with An. gambiae. It looks from the haplotype network like coluzzi Hap_1 might be the same as gambiae Hap_3. Is it? If so, it would aid the reader to change the names so that identical haplotypes in both species have the same name.

Minor issues:

L37. Change ‘a number’ to ‘the number’

L40. Surely IRS is also important as well as ITNs?

L50. Reference [9] is very, very old. Surely there is a more recent appropriate reference than one from 2004?

L52-53. Here, it needs clarifying that P450 mediated resistance is the major mechanism for pyrethroid metabolism. Whilst they have been implicated in carbamate metabolism other insecticide classes are targeted by e.g. GSTs or esterases. So, please ensure the text is clear that you are talking about pyrethroids here.

L63. Needs a ref to pyrethroid metabolisers.

L64. Change pyrethroids type I and II to type I and II pyrethroids.

L67. Insert ‘The’ before ‘CYP6P4’

L68. Change ‘resistance gene’  ‘resistance-associated gene’. I would advocate that CYP6P4 does not ‘explain the resistance’ (indeed it is not clear what is explaining the resistance – SNPs or over-expression?) but variation/over-expression may explain some of the resistance. Please modify.

L69-70. Clarify that this sentence (ref 20) is pertaining to gambiae (given previous sentence was arabiensis).

L96. Change is to are.

L124. Change concentration to concentrations

L126. Change cytochrome P450s to cytochromes P450

L147. Insert ‘the’ before An. gambiae.

L149. Insert ‘both’ before An. coluzzi

L154. Change describes to described

L155. 80x primer/probe mix? Please check. Surely, the reaction is run at 1x so perhaps you need to add a volume here?

L157 on. Do you need to give primer/probe sequences? These are available in the original papers so unless you have modified these why give them?

L172. Subscript 20.

L173. Remove [ ] around reference.

L188. Change comprises to comprised.

L189. Subscript 2 in H2O

L201. Change 1kb upstream region to The region 1kb upstream

L224. Italicise An. gambiae

L231. Change 3th to 3rd

L234. Change the both to both the

L240. Subscript 2 in H2O

L241. Change assay to assays

L264. Insert found exclusively before in Nkolondom

L271. Change P450s to P450

L275-276. Change permethrin 5x to 5x permethrin (or better give actual percentage – see above)

Figure 1 legend. Change permethrin 5x to 5x permethrin (or better give actual percentage – see above)

L293. Change transcription to expression

L302. Insert the before An. coluzzi

L316. Change to ‘Association of polymorphisms in Cyp6m2 and pyrethroid resistance

L318-329. This is not very clear – especially given that further polymorphisms are introduced in the next section on polymorphism analysis.

L319. Insert a before model

L321. The G/T mutation is mentioned as if we know what mutation this is but it is the first mention so it needs explaining.

L325. Change carries to carriers (x2)

L376. Insert the before Haplotype

L379-380. Whilst I am sure that the ‘original’ haplotype was susceptible you cannot say from this very limited sampling anything about the derivation of these haplotypes so remove this sentence.

L389. I am unclear what is meant here with the note about 2n*. 2n* is not used in the table so what bits of this tabulated data are from unphased data?

L395. Do not superscript 392

L417. What is an A/T restriction site? Restriction sites are not 1bp

L453. Insert the before lowest

L454. Change were to was

Table 4. For nucleotide diversity use period (.) not comma (,) as separator.

L502. Do not superscript 168.

L506. Change sample to samples

Table 5. Change in CYP6M2 upstream to upstream of CYP6M2 (and same for P4)

L524. Delete enzyme

L549. delete the

L571. Do not superscript 392

L582. Do not superscript 168.

L621. Change keys to key

Author Response

Mosquito collection (L110 onwards). How was sampling of families avoided? By collecting in many different breeding sites all over the collection area and the larvae are pooled before emergence. The obtained adults are then surely diversified.

L207. ‘known binding sites’. In what species? This is important for accurately detecting such sequences. Added: eukaryotic sequences, with a focus on human and mouse.  

L250. I am unclear what is meant by continuous variables here since mortality in WHO bioassays is binary – dead or alive. So what is the continuous variable? This was an error; it has been removed.

L273-277 talks about tests with 5x and 10x concentrations of insecticide but this is not mentioned in the methods. Please do so. Rather than saying 5x and 10x give actual concentration. If 0.75% is standard then presumably 3.75% and 7.5%. done: the values have been introduced line 125-130 and the text corrected accordingly in lines 273-277.

Figure 2. Using R and S for alleles is extremely confusing in this context since you use R and S also for phenotype. For alleles, use Ser and Phe (or S and F) to avoid confusion. Ensure that this is done throughout the ms. ok

Figure 2 also talks about hybrids but there is no information in the methods that any hybrids were generated. An. gambiae was always crossed with gambiae and coluzzi with coluzzi. This needs attending to or explaining better. This was a bad use of the term “hybrid” and it has been removed.

Section 3.4.1 would benefit from a figure to depict where the polymorphisms being mentioned are with respect to the TSS. All polymorphisms should be described with rs numbers (where available) or genome positions where not. They are presented in supp Table S4 and supp FigS1

Figure 3. These colours need changing. They are difficult to discriminate – particularly for people with colour blindness. We changed to the main colours red, blue, and green.

Somewhere in the text it would be useful to mention whether any An. coluzzi haplotypes are shared with An. gambiae. It looks from the haplotype network like coluzzi Hap_1 might be the same as gambiae Hap_3. Is it? If so, it would aid the reader to change the names so that identical haplotypes in both species have the same name. The main haplotypes are shared for the promoter regions in both species, but the correspondence is weak in the coding region. This was added in the result section Lines 415 and 468. 

Minor issues:

L37. Change ‘a number’ to ‘the number’ ok

L40. Surely IRS is also important as well as ITNs? Added

L50. Reference [9] is very, very old. Surely there is a more recent appropriate reference than one from 2004? Done: (Ranson 2016 added), first characterisation is Hemingway 2004, reason way we keep it.

L52-53. Here, it needs clarifying that P450 mediated resistance is the major mechanism for pyrethroid metabolism. Whilst they have been implicated in carbamate metabolism other insecticide classes are targeted by e.g. GSTs or esterases. So, please ensure the text is clear that you are talking about pyrethroids here. Done

L63. Needs a ref to pyrethroid metabolisers. Added: Adolfi 2019 (6M2), Ibrahim 2016 (6P4)

L64. Change pyrethroids type I and II to type I and II pyrethroids. ok

L67. Insert ‘The’ before ‘CYP6P4’ ok

L68. Change ‘resistance gene’  ‘resistance-associated gene’. I would advocate that CYP6P4 does not ‘explain the resistance’ (indeed it is not clear what is explaining the resistance – SNPs or over-expression?) but variation/over-expression may explain some of the resistance. Please modify. ok

L69-70. Clarify that this sentence (ref 20) is pertaining to gambiae (given previous sentence was arabiensis). Done

L96. Change is to are. okL124. Change concentration to concentrations ok

L126. Change cytochrome P450s to cytochromes P450 ok

L147. Insert ‘the’ before An. gambiae. ok

L149. Insert ‘both’ before An. coluzzii ok

L154. Change describes to described ok

L155. 80x primer/probe mix? Please check. Surely, the reaction is run at 1x so perhaps you need to add a volume here? Corrected, with thanks

L157 on. Do you need to give primer/probe sequences? These are available in the original papers so unless you have modified these why give them? Ok

L172. Subscript 20. Done

L173. Remove [ ] around reference. Done

L188. Change comprises to comprised. Ok

L189. Subscript 2 in H2O MgCl2 indeed.

L201. Change 1kb upstream region to The region 1kb upstream ok

L224. Italicise An. gambiae  Ok

L231. Change 3th to 3rd ok

L234. Change the both to both the ok

L240. Subscript 2 in H2O ok

L241. Change assay to assays ok

L264. Insert found exclusively before in Nkolondom ok

L271. Change P450s to P450 ok

L275-276. Change permethrin 5x to 5x permethrin (or better give actual percentage – see above) Done

Figure 1 legend. Change permethrin 5x to 5x permethrin (or better give actual percentage – see above) Done, percentages added

L293. Change transcription to expression ok

L302. Insert the before An. coluzzii ok

L316. Change to ‘Association of polymorphisms in Cyp6m2 and pyrethroid resistance Done

L318-329. This is not very clear – especially given that further polymorphisms are introduced in the next section on polymorphism analysis. We merged both parts to have a continuous paragraph on the polymorphism of the upstream region.

L319. Insert a before model ok

L321. The G/T mutation is mentioned as if we know what mutation this is but it is the first mention so it needs explaining. Done

L325. Change carries to carriers (x2) ok

L376. Insert the before Haplotype ok

L379-380. Whilst I am sure that the ‘original’ haplotype was susceptible you cannot say from this very limited sampling anything about the derivation of these haplotypes so remove this sentence. Sentence modified

L389. I am unclear what is meant here with the note about 2n*. 2n* is not used in the table so what bits of this tabulated data are from unphased data? Removed, this was to clearly show that this number is for the unphased sequences

L395. Do not superscript 392 ok

L417. What is an A/T restriction site? Restriction sites are not 1bp

L453. Insert the before lowest ok

L454. Change were to was ok

Table 4. For nucleotide diversity use period (.) not comma (,) as separator. ok

L502. Do not superscript 168. ok

L506. Change sample to samples ok

Table 5. Change in CYP6M2 upstream to upstream of CYP6M2 (and same for P4) ok

L524. Delete enzyme ok

L549. delete the ok

L571. Do not superscript 392 ok

L582. Do not superscript 168. ok

L621. Change keys to key ok

Round 2

Reviewer 1 Report

It can be accepted in the present form. 

Reviewer 2 Report

L258. Percentage mortalities not mortalities percentage

L316 and 319 still says hybrid

L392. Derived not derive

L432. What is an A/T restriction site? Restriction sites are not 1bp. (I asked this originally and it has not been addressed)